# First Characterization of ADAMTS-4 in Kidney Tissue and Plasma of Patients with Chronic Kidney Disease—A Potential Novel Diagnostic Indicator

**DOI:** 10.3390/diagnostics12030648

**Published:** 2022-03-07

**Authors:** Ivana Kovacevic Vojtusek, Mario Laganovic, Marija Burek Kamenaric, Stela Bulimbasic, Stela Hrkac, Grgur Salai, Vanja Ivkovic, Marijana Coric, Rudjer Novak, Lovorka Grgurevic

**Affiliations:** 1Department of Nephrology, Arterial Hypertension, Dialysis and Transplantation, University Hospital Center Zagreb, 10000 Zagreb, Croatia; ikovacevicvojtusek@gmail.com (I.K.V.); vanja.ivkovic@gmail.com (V.I.); 2Department of Nephrology, University Hospital Merkur, 10000 Zagreb, Croatia; mlaganovic@gmail.com; 3Tissue Typing Center, Clinical Department for Transfusion Medicine and Transplantation Biology, University Hospital Center Zagreb, 10000 Zagreb, Croatia; mburek@kbc-zagreb.hr; 4Clinical Department of Pathology and Cytology, University Hospital Center Zagreb, 10000 Zagreb, Croatia; stela.bulimbasic@gmail.com (S.B.); marijanacoric17@gmail.com (M.C.); 5Department of Emergency Medicine, University Hospital Center Zagreb, 10000 Zagreb, Croatia; stela.hrkac@gmail.com; 6Center for Translational and Clinical Research, Department of Proteomics, School of Medicine, University of Zagreb, 10000 Zagreb, Croatia; salai.grgur@gmail.com (G.S.); rudjer.novak@gmail.com (R.N.); 7Teaching Institute of Emergency Medicine of the City of Zagreb, 10000 Zagreb, Croatia; 8Department of Public Health, Faculty of Health Studies, University of Rijeka, 51000 Rijeka, Croatia; 9Department of Anatomy, School of Medicine, University of Zagreb, 10000 Zagreb, Croatia

**Keywords:** ADAMTS-4, bone morphogenic protein 1, chronic kidney disease, kidney transplantation, renal dialysis

## Abstract

Background: We have previously shown that metzincin protease ADAMTS-4 accompanies renal fibrogenesis, as it appears in the blood of hemodialysis patients. Methods: Native kidney (NKB) and kidney transplant (TXCI) biopsy samples as well as plasma from patients with various stages of CKD were compared to controls. In paired analysis, 15 TXCI samples were compared with their zero-time biopsies (TX0). Tissues were evaluated and scored (interstitial fibrosis and tubular atrophy (IFTA) for NKB and Banff ci for TXCI). Immunohistochemical (IHC) staining for ADAMTS-4 and BMP-1 was performed. Plasma ADAMTS-4 was detected using ELISA. Results: ADAMTS-4 IHC expression was significantly higher in interstitial compartment (INT) of NKB and TXCI group in peritubular capillaries (PTC) and interstitial stroma (INT). Patients with higher stages of interstitial fibrosis (ci > 1 and IFTA > 1) expressed ADAMTS-4 in INT more frequently in both groups (*p* = 0.005; *p* = 0.013; respectively). In paired comparison, TXCI samples expressed ADAMTS-4 in INT and PTC more often than TX0. ADAMTS-4 plasma concentration varied significantly across CKD stages, being highest in CKD 2 and 3 compared to other groups (*p* = 0.0064). Hemodialysis patients had higher concentrations of ADAMTS-4 compared to peritoneal dialysis (*p* < 0.00001). Conclusion: ADAMTS-4 might have a significant role in CKD as a potential novel diagnostic indicator.

## 1. Introduction

Chronic kidney disease (CKD) is a heterogeneous disorder with multiple causes that endangers the structure and function of the kidney. Its occurrence is on a steady rise since the 1990s and has recently reached a global age-standardized prevalence of 9.1% [1]. It is widespread in patients with diabetes, hypertension, obesity and atherosclerosis, affecting around 500 million individuals worldwide [2,3]. CKD is defined by progressive renal deterioration accompanied by loss of function, typically over months or years. Five stages of the disease are recognized based primarily on the patients glomerular filtration rate (GFR) [4]. Stage 5 CKD is also referred to as end-stage renal disease (ESRD), as such patients cannot survive long without some form of renal replacement therapy, such as dialysis or transplantation. To date, there is no approved therapy to halt or reverse the progression of CKD and restore kidney function although early diagnosis and new therapeutic approaches may delay disease progression and the need for renal replacement [5]. The disease develops progressively, starting as small lesions that deteriorate over time and lead to renal fibrosis. Similar to wound healing, renal fibrosis probably initiates as a physiological response to injury. This process is mediated by renal fibroblasts that start proliferating and producing increased amounts of extracellular matrix (ECM). In so doing, they express α-smooth muscle actin, indicating their activation [6,7,8]. As opposed to physiological wound healing, this pathological scarring process continues and becomes a major determinant of CKD progression [9]. Readily available treatment options, such as dialysis and kidney transplantation, pose a variety of risks or side effects to the patient. These include the risk of infection, rejection of organ transplant, and fatigue with treatment, which can ultimately lead to patient mortality. Therefore, there is a clear need for a more advanced, self-healing approach [10].

The regenerative capacity of the nephron and peritubular capillaries is not well understood; however, the existence of a renal stem cell was proposed [11]. The search for a more specific approach in targeting CKD has revealed a plethora of molecules with profibrotic or antifibrotic properties. Our previous research has hinted the importance of a new metalloproteinase, bone morphogenetic protein 1-3 (BMP1-3) as an important mediator. Inhibition of circulating BMP1-3 in an animal CKD model with specific BMP1-3 polyclonal antibodies, as opposed to rats treated with the BMP1-3 protein, increased the overall survival rate by 60% and significantly reduced the level of kidney fibrosis and glomerular sclerosis, indicating a high benefit of the therapeutic intervention [12].

Our previous research has also shown for the first time that metzincin protease family member ADAMTS-4 (also known as aggrecanase-1) appears in the peripheral blood of human patients with ESRD [13]. As ADAMTS-4 processes a wide range of matrix proteoglycans, such as aggrecan, versican, brevican, decorin, neurocan, transferrin, and fibromodulin, it could be important in CKD pathogenesis. In accordance with this hypothesis, Rudnicki et al. associated the expression of renal V0 and V1 isoforms of versican to progressive renal disease [14]. Although the regulation of ADAMTS-4 expression is not fully understood, it is tightly controlled on multiple levels, including transcription, translation, and through physiologic enhancers or inhibitors. Several molecules, such as interleukin 1 (IL-1), tumor necrosis factor (TNF), transforming growth factor beta (TGF-β), IL-17, fibronectin, retinoic acid, and neprilysin, have been identified as major ADAMTS-4 enhancers [15]. Inflammation is also a potent stimulus: renal injury induces the release of chemotactic factors and infiltration of macrophages that readily polarize to the M1 pro-inflammatory phenotype. The M1 macrophages secrete inflammatory cytokines, and if unopposed, this may lead to tissue damage, fibrosis, vascular changes, and inflammation, at the same time suppressing the M2 tissue-repair macrophage polarization [16]. ADAMTS-4 expression has also been confirmed in macrophage-rich areas of atherosclerotic plaques, and its expression is augmented during atherogenesis [17]. 

Discovery of a biomarker that could identify patients with early-stage CKD is of great importance, as it could lead to timely clinical interventions and more favorable patient outcomes. Thus far, there is only a limited number of papers reflecting on the role of ADAMTS-4 in kidney biology. This small-scale study strengthens the hypothesis that the ECM modulator ADAMTS-4 has a role in human CKD. Lessons learned about its role in the atherosclerotic milieu could prove valuable for decoding the intricate pathways ADAMTS-4 modulates in CKD. Therefore, we investigated the changes in expression and localization of ADAMTS-4 in the kidney (i.e., its compartments) and also the changes in concentration of ADAMTS-4 released to the plasma of patients with different stages of CKD. Expression levels in renal tissue compartments were correlated to CKD progression, and changes of its plasma concentrations were monitored across the same groups. 

## 2. Materials and Methods

### 2.1. Sample Collection and Study Outline

This prospective observational study was approved by the Ethics Committee of the University Hospital Center Zagreb (EP-16/106-2). All participants provided a signed informed consent. Blood and tissue samples were collected from patients between January 2018 and June 2019 at the Department of Nephrology, Arterial Hypertension, Dialysis and Transplantation, UHC Zagreb. 

Blood samples were drawn into tubes with anticoagulant substance containing 3.8% sodium citrate to form an anticoagulant-to blood ratio (*v*/*v*) 1:9. Plasma was obtained by centrifugation (15 min at 4 °C and 3000× *g*), and aliquots of each sample were stored at −80 °C until analysis. Experimental groups included: 32 patients in CKD stages 1–5, 5 hemodialysis (HD), 7 peritoneal dialysis (PD) patients, 30 kidney transplant patients (TX), and a control group of 5 healthy volunteers (CTRLpl) (Figure 1).

Kidney biopsy tissue samples were obtained from patients with CKD: 19 patients who underwent native kidney biopsy for clinical diagnosis of kidney disease (NKB group: 9 in CKD stages 1–2 and 10 in CKD stages 3–5) and 34 patients at different timepoints after kidney transplantation (TXCI group), who underwent kidney transplant biopsy for clinical evaluation (impairment of kidney function or proteinuria). As controls (CTRL group), 8 cadaveric kidney donors without evidence of CKD (eGFR > 90 mL/min, absence of proteinuria or any structural changes), who underwent transplant nephrectomies, were used (Figure 1). In 15 subjects from the TXCI group “zero-time”, donor kidney biopsy was performed at the day of transplantation (TX0). These samples were obtained from the hospital archive and used for paired analyses (15 TXCI-TX0 pairs) (Figure 1).

Kidney tissue samples biopsies underwent immunohistochemical (IHC) staining for ADAMTS-4 and BMP-1. Histopathology and immunohistochemistry analyses were performed in Clinical Department for Pathology and Cytology of University Hospital Center Zagreb. 

All patient clinical data were obtained from the hospital information system, and clinical data of kidney transplant donors for zero-time kidney biopsies were taken from kidney donor reports.

### 2.2. Kidney Biopsy Sample Collection and Analysis

Native kidney biopsy was performed by standard procedures [18] using ultrasonography and biopsy needle guidance technique. Histologic evaluation by light microscopy and immunofluorescence were performed routinely, combined with electron microscopy in some cases. Interstitial fibrosis and tubular atrophy (IFTA) scores (0–3) were used for the assessment of the chronicity of kidney disease in NKB group. The Banff Working Classification was used for estimation of pathology of kidney transplant biopsy samples (TXCI, TX0) [19]. 

### 2.3. Immunohistochemistry Analysis (IHC)

Tissue ADAMTS-4 and BMP-1 expression levels were determined on formalin-fixed, paraffin embedded (FFPE) tissue sections by IHC. Selected tissues were cut into 4 µm sections, placed on charged slides, and dried at 60 °C for one hour. Slides were cooled to room temperature, deparaffinized with Clarify (American Mastertech, Lodi, CA, USA), and rinsed in water in a Dako Omnis platform (Agilent Technologies, Santa Clara, CA, USA). Epitopes were retrieved with treatment with Dako Flex TRS Low (Agilent Technologies, Santa Clara, CA, USA) for 30 min at 97 °C. Slides were incubated with primary rabbit polyclonal ADAMTS-4 antibody (Abcam, Cambridge, MA, USA) at a dilution 1:200 for 60 min, followed by Dako GV800 HRP-conjugated secondary antibody (Agilent Technologies, Santa Clara, CA, USA) for 20 min. For BMP-1 expression, detection slides were incubated with primary rabbit polyclonal antibody against BMP-1 (HPA014572; Sigma-Aldrich, Merck, Germany) at a dilution 1:50 for 60 min. Thereafter, primary antibody signal amplifier (Rabbit Linker Dako GV809 (Agilent Technologies, Santa Clara, CA, USA)) was applied for 10 min, followed by Dako GV800 HRP-conjugated secondary antibody (Agilent Technologies, Santa Clara, CA, USA) for 20 min. 3,3′-diaminobenzidine chromogen (Agilent Technologies, CA) was applied for 5 min, contrasted with hematoxylin, and cover-slipped. Stained slides were analyzed using an optical microscope (Zeiss Axiostar plus, Oberkochen, Germany; magnification range 200× and 400×). As positive control for ADAMTS-4, middle-sized intralobular artery in kidney biopsy sample was used. ADAMTS-4 was IHC positive in artery wall in the same pattern as previously described by Dong et al. [20]. 

IHC analyses were performed in three main kidney tissue compartments: (*a) interstitial*—peritubular capillaries (PTC) and interstitial stroma (INT), (*b) glomerular*—glomerular capillaries (GC) and Bowman space (BW), (*c) tubules*—proximal (PXT) and distal (DT). For both molecules (ADAMTS-4 and BMP-1), staining was defined as 0—negative or 1—positive for each of the six separate observation areas (PTC, INT, GC, BW, PXT, DT). Positive staining was defined if staining was present in more than 1% of observed area [21]. IHC evaluation was performed independently by two board certified nephropathologists (SB, MC) that were blinded to group allocation.

### 2.4. Enzyme-Linked Immunosorbent Assay (ELISA)

ADAMTS-4 in plasma samples was detected using an indirect ELISA kit (Human ADAMTS-4 DuoSet ELISA DY4307-05 R&D, Minneapolis, MN, USA) according to manufacturer’s instructions. Results were obtained with a plate reader (Molecular Devices–SpectraMax i3x) at 450 nm. All samples and standards were analyzed in duplicates and samples with an individual coefficient of variation (CV) greater than 25% were retested in duplicates. All procedures and evaluation of the results were conducted by researchers blinded to clinical and pathological patient data. 

### 2.5. Statistical Analysis

Normality of distribution of continuous variables was tested using D’Agostino-Pearson test. Categorical variables were presented as absolute numbers and percentages. Normally distributed continuous variables were presented as mean and standard deviation and non-normally distributed as median and interquartile range. The frequencies of ADAMTS-4 and BMP-1 proteins in the examined kidney biopsy samples of CTRL, NKB and TXCI group as well as for TX0-TXCI sample pairs were obtained by direct counting based on the positive/negative immunohistochemistry staining. Data distribution differences were compared using Fisher’s exact test with Yates correction. Plasma ADAMTS-4 concentration in the group of 8 patients with CKD was presented as multiple variables graphs with box-and-whisker plot for median. The difference in means between the two groups were analyzed by Student’s *t*-test and ANOVA One-Way Analysis of Variance test for more than two independent groups. A two-tailed *p*-value < 0.05 was used as indicator of statistical significance. All analyses were carried out using MedCalc software (version 19.2.6).

## 3. Results

### 3.1. Immunohistochemistry Analysis of ADAMTS-4 and BMP-1 Molecules in Kidney Biopsy Samples of Patients with CKD

Immunohistochemistry analysis included kidney tissue biopsy samples from 19 patients who underwent native kidney biopsy (NKB), 34 patients with transplanted kidney who underwent biopsy due to a clinical indication (TXCI), and 8 cadaveric kidney donors who underwent transplant nephrectomies (CTRL). Study group characteristics are presented in Table 1.

### 3.2. Distribution of ADAMTS-4 and BMP-1 in Kidney Tissue Samples of CTRL, NKB and TXCI Group

Distribution of patient frequencies for ADAMTS-4 and BMP-1 IHC staining in their kidney biopsy samples (positive IHC staining frequency is presented) in each of six separate observation areas (PTC, INT, GC, BW, PXT, and DT) for CTRL and groups with chronic kidney disease (NKB and TXCI groups, both separately presented as CKD 1–2 or CKD 3–5 stage) is shown in Figure 2.

#### 3.2.1. BMP-1 and ADAMTS-4 in Kidney Samples without CKD

BMP-1 was present in proximal tubules of 88% and glomerular Bowman space of 37% subjects in CTRL group and was totally absent in interstitial compartment and glomerular capillaries. ADAMTS-4 was present in tubular (both in PXT and DT) and glomerular (both in GC and BW) compartments of samples without CKD. Only 13% of subjects expressed ADAMTS-4 in interstitial stroma and none in peritubular capillaries. 

#### 3.2.2. BMP-1 and ADAMTS-4 in Kidney Samples with CKD

In CKD, BMP-1 was also present only in tubular compartment (PXT and DT) and BW in both study groups, NKB and TXCI, respectively. Frequencies of BMP-1 IHC staining in tubules were not significantly different between CTRL and groups with CKD, NKB, or TXCI, respectively (CTRL vs. NKB, 87.5% vs. 94.7%, *p* = 1 and CTRL vs. TXCI, 87.5% vs. 97%, *p* = 0.348 for PXT, respectively, CTRL vs. NKB, 0% vs. 31.5%, *p* = 0.279 and CTRL vs. TXCI, 0% vs. 14.7%, *p* = 0.563; for DT, respectively). Similar was determined for BW (CTRL vs. NKB, 37.5% vs. 15.8%, *p* = 0.319 and CTRL vs. TXCI, 37.5% vs. 20.5%, *p* = 0.369, respectively). Even when we compared CKD subgroups (CKD 1–2 or CKD 3–5) with CTRL for each area, frequencies of BMP-1 positivity were similar (CTRL vs. NKB-CKD 1–2, CTRL vs. NKB-CKD 3–5, CTRL vs. TXCI-CKD 1–2, and CTRL vs. TXCI 3–5; all *p* > 0.05), with only one exception for DT area, where frequency of BMP-1 staining was significantly higher in NKB-CKD 1–2 than in CTRL (0% vs. 47%, *p* = 0.035). Additionally, the CKD subgroups (CKD 1–2 and 3–5) from both groups (NKB or TXCI) were not different among each other regarding frequency of patients positivity for BMP-1 in each of the above-mentioned areas (*p* > 0.05, respectively).

In CKD, ADAMTS-4 IHC staining was detected in all six observation areas. When comparing frequencies of patients, regarding ADAMTS-4 IHC staining positivity between the CTRL and groups with CKD (NKB or TXCI) for all areas, statistically significant difference was found only in interstitial compartment (CTRL vs. NKB, 0% vs. 52.6%, *p* = 0.025; CTRL vs. TXCI, 0% vs. 97%, *p* < 0.001 for PTC, respectively, and CTRL vs. NKB, 14.3% vs. 57.9%, *p* = 0.043; CTRL vs. TXCI, 14.3% vs. 64.7%, *p* = 0.015 for INT, respectively) and glomerular capillaries (CTRL vs. NKB, 37.5% vs. 100%, *p* = 0.001; CTRL vs. TXCI, 37.5% vs. 85.3%, *p* = 0.012 for GC, respectively), being significantly higher in both groups with CKD (NKB or TXCI). 

When comparing ADAMTS-4 expression between CKD subgroups (CKD 1–2 vs. CKD 3–5) within NKB or TXCI, significant difference was found only in PTC area of NKB group (CKD 1–2 vs. CKD 3–5; 5.3% vs. 47.4%; *p* = 0.001) and was also marginally significant in INT (NKB-CKD 1–2 vs. NKB-CKD 3–5; 15.8% vs. 42.1%, respectively; *p* = 0.069), both being higher in advanced stage of CKD (CKD 3–5). The same was not observed within TXCI group.

Interestingly, when we compared among equivalent CKD subgroups of NKB and TXCI, similar ADAMTS-4 expression frequency was found in all six observed areas (NKB-CKD 1–2 vs. TXCI-CKD 1–2; NKB-CKD 3–5 vs. TXCI-CKD 3–5; all *p* > 0.05 for PTC, INT, PXT, DT, GC, and BW, respectively). 

ADAMTS-4 and BMP-1 IHC expression in kidney biopsy samples from NKB group is presented in Figure 3.

### 3.3. Association of ADAMTS-4 Expression in Interstitial Compartment with Chronic Kidney Histology Scores

To further elucidate potential correlation of ADAMTS-4 with progression of chronic kidney disease, we correlated ADAMTS-4 IHC staining in interstitial compartment of kidney tissue (PTC and INT) with chronic kidney tissue interstitial fibrosis scores: chronic interstitial score (ci) per Banff classification for TXCI group and IFTA stage for NKB group. When dichotomizing TXCI group by ci score as ci ≤1 or >1, significantly more patients with ci > 1 expressed ADAMTS-4 in INT area (ci > 1 vs. ci ≤ 1; 100% vs. 47.8%, respectively; *p* = 0.005), which was not observed for PTCs. Similarly, when dichotomizing NKB group by IFTA stage as IFTA ≤ 1 or IFTA > 1, significantly more patients with IFTA > 1 expressed ADAMTS-4 in INT (IFTA > 1 vs. IFTA ≤ 1; 100% vs. 33.3%, respectively; *p* = 0.013) and in PTCs (IFTA > 1 vs. IFTA ≤ 1; 100% vs. 25%, respectively; *p* = 0.003).

As an additional unexpected observation, ADAMTS-4 IHC expression in proximal tubules was found to be associated with presence of acute tubular injury (ATI) in native and transplant kidney biopsy samples. ADAMTS-4 expression in PXT was significantly more frequent in patients with ATI than without in NKB (100% vs. 7.2%, respectively; *p* = 0.001) and TXCI group (87.5% vs. 0%, respectively; *p* < 0.001). Such association was not found for the BMP-1 molecule (NKB 100% vs. 92.8%, respectively; TXCI 100% vs. 96.1%, respectively; for all *p* > 0.05).

### 3.4. Analysis of ADAMTS-4 IHC Expression in 15 Transplant Kidney Biopsy Sample Pairs (TX0-TXCI)

The results of a paired sub-analysis comparing 15 kidney biopsy samples from TXCI group performed at different timepoints after transplantation with their paired (same allograft) “zero time “samples (TX0) performed at the day of transplantation (15 TX0-TXCI pairs) are presented in Table 2.

In paired analysis, subjects in TXCI group had significantly higher CKD stage and ci score than in TX0 group (TXCI vs. TX0, all *p* < 0.001, Table 2) Concordantly, significantly higher number of biopsy samples in TXCI compared to TX0 group expressed ADAMTS-4 in both observed areas of interstitial compartment (10/15 vs. 1/15, respectively; *p* = 0.00169 in INT, 15/15 vs. 0/15, respectively; *p* < 0.0001 for PTCs) and glomerular capillaries (13/15 vs. 5/15, respectively; *p* = 0.00778). 

ADAMTS-4 expression by IHC staining for one pair of TXCI-TX0 biopsy samples is presented in Figure 4.

### 3.5. Detection of ADAMTS-4 in Plasma of Patients with CKD (Elisa Analysis)

Distribution of ADAMTS-4 concentration in plasma of the group of eight patients with CKD (CKD stages 1–5, PD, HD before and after procedure: HD-B and HD-A, TX) and controls (CTRLpl) is presented in Figure 5. When comparing mean ADAMTS-4 plasma concentrations between CTRLpl and all studied groups (CKD 1–5, PD, HD, TX), significant difference was found between CTRLpl and HD group (CTRL vs. HD-B; 226 vs. 1032.6; *p* = 0.006). ADAMTS-4 plasma concentration varied significantly between CKD stages, being highest in CKD stages 2 and 3, taken together and compared to all other CKD stages (CKD 2 + 3 vs. 1 + 4 + 5; 2925.1 vs. 613.6; respectively *p* = 0.0064) or to all studied groups taken together (CKD 2 + 3 vs. CKD1 + 4 + 5 + HD-B + PD + TX; *p* = 0.00004) Interestingly, when analyzing only dialysis groups, none of the patients in PD group had detectable ADAMTS-4 in plasma, which differed significantly from the HD group (PD vs. HD-B; 0 vs. 1032.6, respectively; *p* < 0.00001). Additionally, ADAMTS-4 plasma values in HD group were not different even after a regular 4 h hemodialysis procedure (HD-B vs. HD-A; 1032.6 vs. 1171, respectively; *p* = 0.084).

## 4. Discussion

In this study, to the best of our knowledge, for the first time, ADAMTS-4 was characterized in kidney tissues and plasma of patients with CKD. Previous research of ADAMTS-4 in kidney diseases is scarce. Wang et al. reported an elevated ADAMTS-4 expression in patients with early stage membranous nephropathy; however, because of the use of tissue lysates, the exact origin of the protein could not be determined [22]. A single animal study indicated that ADAMTS-4 expression in healthy murine kidneys occurs in proximal and distal convoluted tubules [23]. We have determined that patients of the native kidney biopsy group (NKB) with early-stage CKD predominantly express ADAMTS-4 in the glomerular and tubular and only slightly in the interstitial compartment. Taking into account that almost 90% of our NKB group was diagnosed with primary glomerular disease, our results concur with the Wang group. 

The underlying pathology of CKD and ESRD is renal fibrosis, which initiates mainly in the peritubular capillaries of the interstitial compartment [9,24]. Other ADAMTS proteins have previously been implicated in the pathogenesis of kidney fibrosis: higher expression of ADAMTS-1 was detected in the early course of fibrogenesis on mouse model of unilateral ureteral obstruction. Investigators defined endothelial cell of PTC as the source of ADAMTS-1 in the process of pericytes detachment and PTC destabilization [25]. ADAMTS-2 and -12 were upregulated in mesenchymal interstitial cells during their transformation to myofibroblasts [26]. Furthermore, ADAMTS-1, -12, and -15 were found upregulated in animal model of Adriamycin-induced kidney fibrosis [27]. Our results implicate ADAMTS-4 in the progression of CKD and thus possibly the pathogenesis of kidney fibrosis. Indeed, there is no clear evidence on the mechanism of action of ADAMTS-4 in renal failure although its function may be assumed by studying its role in other organ systems. Boyd et al. showed that fibroblasts produce ADAMTS-4 in response to tissue-damage, which leads to ECM remodeling and promotion of immune cell infiltration in the lungs [28]. Additionally, the circulating active form of the protein may play a role in the regulation of ECM expression and deposition, ultimately leading to fibrosis and a disease state. ADAMTS-4 expression in the interstitial compartment was elevated across all disease groups; however, it was most abundantly expressed in non-transplanted patients with advanced disease stages (CKD 3–5). Similar localization of ADAMTS-4 in the interstitial compartment among equivalent CKD stages of NKB and TXCI groups implicates a similar role of ADAMTS-4 in CKD progressivity. Furthermore, increased expression of ADAMTS-4 in the interstitial compartment significantly correlated to higher chronic interstitial fibrosis scores (IFTA for NKB and ci for TXCI). Finally, paired consecutive biopsy samples from the transplant group clearly confirmed that ADAMTS-4 expression rises proportionally to the amount of interstitial fibrosis (ci score) and stage of CKD. ADAMTS-4 expression was also detected in distal tubules of almost all biopsy samples regardless of presence of CKD and to significantly lesser extent in proximal tubules, whereas its expression was significantly associated with the presence of acute tubular injury. 

These effects might be partially explained by the role ADAMTS-4 has in the processing of extracellular matrix [29]. Expression of versican, a key ADAMTS-4 substrate has been associated to CKD progression in animal models and confirmed in patients with proteinuric kidney diseases [14,30,31]. Inflammatory cells, namely macrophages, seem to be important in the early phase of CKD and kidney fibrosis [26,32,33]. This pro-inflammatory milieu drives the expression of ADAMTS-4 enzymes by macrophages and macrophage-like cells derived from vascular smooth muscle cells. A positive feedback loop is supported where its substrates versican and aggrecan enhance plaque formation, thus fueling fibrosis. The cleavage of these proteoglycans by ADAMTS-4 in turn further destabilizes the atherosclerotic plaques, leading to unfavorable patient outcomes [34,35]. The mechanisms of action of ADAMTS-4 proposed in atherosclerosis might also be relevant in CKD; however, this remains to be elucidated [15,17,20]. We have shown that ADAMTS-4 expression is enhanced in the glomerular capillaries of patients with CKD for both study groups (NKB and TXCI) as compared to controls. The significance of such correlation is yet to be explored, but presumably, ADAMTS-4 might be expressed by activated circulating macrophages as a part of kidney disease. To further assess the role of ADAMTS4 in CKD fibrosis, we also tested the expression of BMP-1, a tolloid-like proteinase, originally identified for its roles in the maturation of procollagen [36]. Both molecules, BMP-1 and ADAMTS-4, were detected in different tubular compartments. Although previous studies found that early inhibition of BMP1–3 significantly reduced kidney fibrosis in rats [12,37], we saw no correlation of BMP-1 expression to CKD progression. 

The analysis of plasma ADAMTS-4 revealed its expression across all stages of CKD, however, with significant variations. As the disease progresses, the expression levels peak at CKD stages 2 and 3 and afterwards sharply decrease towards ESRD. It could be speculated that higher systemic concentrations of ADAMTS-4 might be the result of inflammation that underlies kidney diseases. This is especially the case in primary glomerular disease; however, larger studies are needed for any further conclusions. In renal replacement therapies, ADAMTS-4 was detected in patients on hemodialysis before but also after dialysis. This finding is in concordance with previous research and could be explained by the fact that ADAMTS-4 is a large uremic toxin, which is poorly filtrated using a standard dialysis filter [13]. Interestingly, none of the patients on peritoneal dialysis had detectable plasma ADAMTS-4. As peritoneal dialysis has different capacity for large molecule excretion, it can be assumed that ADAMTS-4 could be excreted using this treatment, but this should be additionally tested. Alternatively, this discrepancy could stem from the fact that hemodialysis patients tend to have more comorbidities, especially cardiovascular diseases and atherosclerosis, which could be independent sources of elevated plasma ADAMTS-4 [38]. That being said, the main limitation of our study is also its relatively small sample size. The presented concepts should be tested further on larger pool of patients with stratified primary kidney disease and various stages of CKD. For similar reasons, as the characterization of ADAMTS-4 in kidney samples from CKD patients was done for the first time by IHC, we opted not to apply intensification of IHC staining, which could be used in further studies. 

## 5. Conclusions

This study characterized ADAMTS-4 proteinases for the first time in kidney tissue samples and plasma of patients with various stages of CKD. Based on its significantly higher IHC expression in peritubular capillaries and interstitial space of patients with CKD compared to controls, its stepwise abundance along with progression of interstitial fibrosis, which was also confirmed on the paired biopsy samples of the 15 transplanted kidneys, ADAMTS-4 might be a novel marker of CKD. Our preliminary results of plasma analysis also confirmed the presence of circulating ADAMTS-4 in CKD patients, being highest in stages 2 and 3, with subsequent disappearance until stage 5, which can fit into the dynamics of the development of the disease. Further prospective studies in larger cohorts are needed to validate the results of the present study.

## Figures and Tables

**Figure 1 diagnostics-12-00648-f001:**
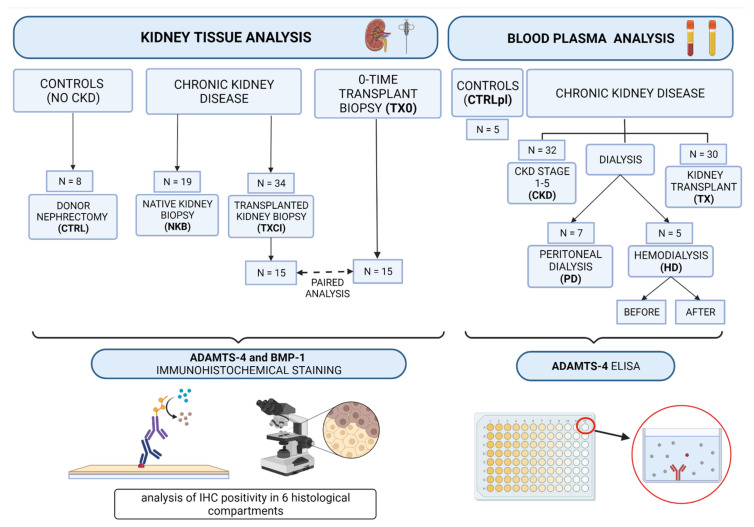
Study outline showing subject groups and the methodological approach.

**Figure 2 diagnostics-12-00648-f002:**
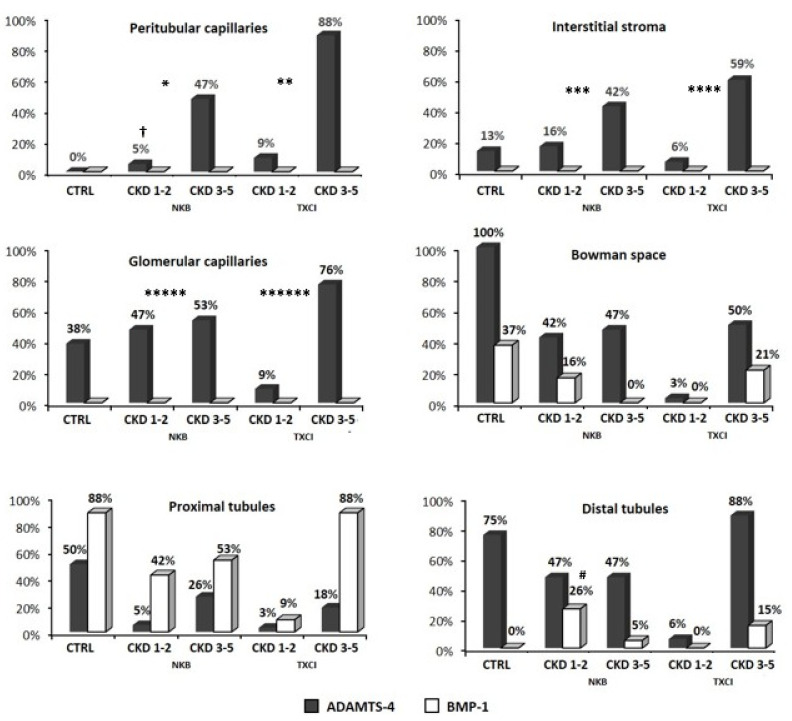
The distribution of ADAMTS-4 and BMP-1 molecules detected by immunohistochemistry staining in six observed areas of different kidney tissue compartments (peritubular capillaries, interstitial stroma, glomerular capillaries, Bowman space, proximal and distal tubules) among control group (CTRL), native kidney biopsy group (NKB), and clinically indicated transplant biopsy group (TXCI). The frequencies of BMP-1 staining between the CTRL and other study groups were not significantly different (all *p* > 0.05), with only one exception for DT area (mark #), where frequency of BMP-1 staining was significantly higher in NKB-CKD 1–2 than in CTRL (*p* = 0.035). ADAMTS-4 IHC staining was detected in all six observation areas. Significantly higher frequency in both group with CKD (NKB and TXCI) in comparison to CTRL group was found for PTC (CTRL vs. NKB, *p* = 0.025, mark *; CTRL vs. TXCI, *p* < 0.001, mark **); for INT (CTRL vs. NKB, *p* = 0.043, mark ***; CTRL vs. TXCI, *p* = 0.015, mark ****); and for GC (CTRL vs. NKB, *p* = 0.001, mark *****; CTRL vs. TXCI, *p* = 0.012, mark ******). When comparing CKD subgroups within NKB and TXCI, significance was found only in NKB group for PTC area (NKB-CKD 1–2 vs. NKB-CKD 3–5; *p* = 0.001, mark †). CKD, chronic kidney disease; CKD 1–2 and CKD 3–5, CKD subgroups of NKB and TXCI group (number represents stage); Frequency (%), proportion of patients in the group who show positive IHC staining for ADAMTS-4 and BMP-1 for 6 different observation areas.

**Figure 3 diagnostics-12-00648-f003:**
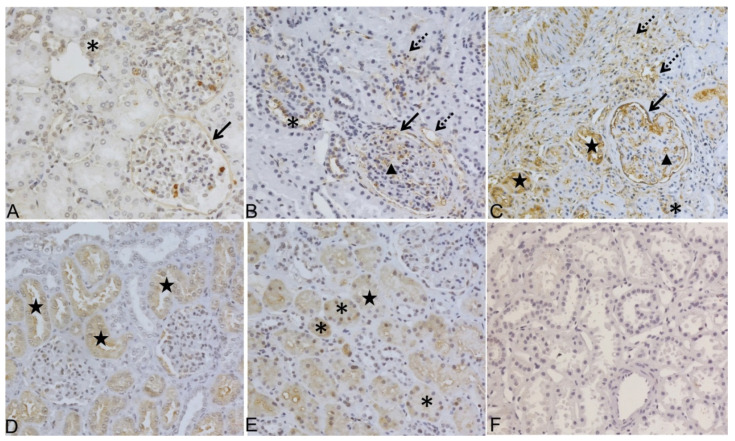
ADAMTS-4 and BMP-1 expression analyses by immunohistochemistry (IHC) in native kidney tissue biopsy samples (NKB) from patients with chronic kidney disease (CKD) in different stages of kidney failure. (**A**) Representative staining of ADAMTS-4 in control without CKD: positive in DT and BC; (**B**) representative staining of ADAMTS-4 in early stage of CKD (Stage 2) positive in GC, peritubular capillaries (PTC), mild fibrosis in interstitial area INT, BC, and DT; (**C**) representative staining of ADAMTS-4 in advanced stages of CKD (Stage 4) positive in all six analyzed kidney areas (GC, PTC, INT, PXT, DT, and BC); (**D**) representative staining of BMP-1 in control without CKD: positive only in PXT and (**E**) in early stage of CKD (Stage 2), positive in DT and PXT with no labelling find in glomerulus; (**F**) secondary antibody only control. Original magnification 200×. DT, asterisk; BC, black arrow; GC, black arrow head; PTC, INT, dashed arrow; PXT, black star.

**Figure 4 diagnostics-12-00648-f004:**
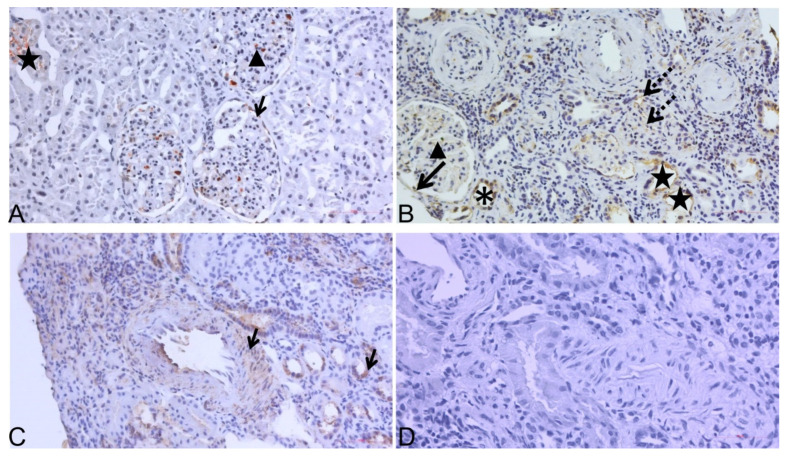
ADAMTS-4 expression analyses by immunohistochemistry (IHC) in transplant kidney biopsy sample pair. (**A**) Representative staining of ADAMTS-4 in “zero-time“ sample (TX0) taken at time of transplantation: positive in GC, BC, and PXT; (**B**) representative staining of ADAMTS-4 in paired TXCI biopsy sample (stage CKD-4): positive in PTC and INT, where interstitial infiltrate is also seen along with GC and damaged PXT; (**C**) ADAMTS-4 positive control: positive in artery wall of middle sized intralobular artery (black arrow—left) and DT (black arrow—right); (**D**) secondary antibody only control in kidney biopsy sample was used. Original magnification 200×. DT, asterisk; BC, black arrow; GC, black arrow head; PTC, INT, dashed arrow; PXT, black star.

**Figure 5 diagnostics-12-00648-f005:**
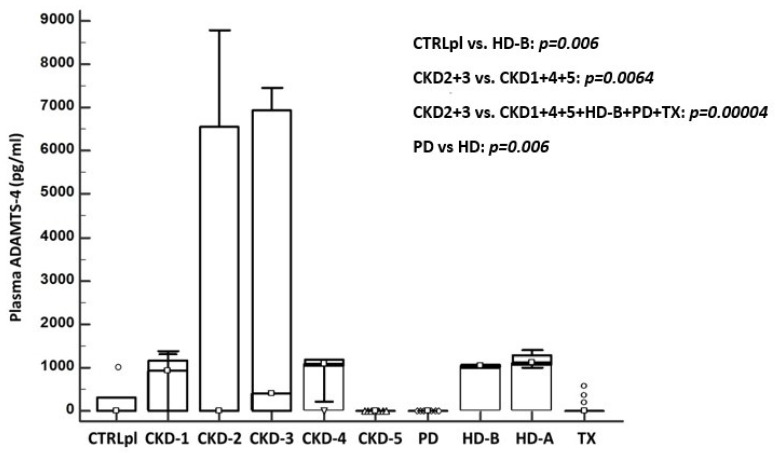
Concentration of ADAMTS-4 protein detected in plasma of healthy subjects in control group (CTRLpl), chronic kidney disease (CKD stages 1–5), on peritoneal dialysis (PD), hemodialysis (before HD-B and after HD-A), and kidney transplant patients (TX). The box plot displays the lower to upper quartile (25th to 75th percentiles) and median (middle line); the minimum and maximum levels as horizontal lines outside the box; and far-out values are plotted with a different marker (◦). Significant difference of mean ADAMTS-4 plasma concentrations was found between CTRLpl and HD-B group (*p* = 0.006) and also varied significantly between CKD stages being highest in patients with CKD stages 2 and 3 compared to CKD 1 + 4 + 5 (*p* = 0.0064) and CKD 1 + 4 + 5 + HD-B + PD + TX (*p* = 0.00004). None of the patients in PD group had detectable plasma ADAMTS-4, which differed significantly from the HD group (*p* < 0.00001).

**Table 1 diagnostics-12-00648-t001:** Study group characteristics for ADAMTS-4 and BMP-1 immunohistochemistry analysis of kidney biopsy tissue samples.

	CTRL N = 8	NKB N = 19	TXCI N = 34	TXCI-TX0 PAIRS N = 15	
				TXCI (N = 15)	TX0 (N = 15)
AGE (YEARS)	53 ± 13	43 ± 17	50 ± 16	51 ± 44	51 ± 7
SEX (% MALE)	2 (25%)	7 (37%)	28 (82%)	14 (93%)	6 (40%)
BMI	24.3 ± 4.6	26.1 ± 5.7	27.1 ± 5.4	27.9 ± 18.5	25.6 ± 2.2
EGFR (ML/MIN)	108 ± 28	56 ± 36	37 ± 16	38 ± 64	91 ± 22
24-PROT G/L	-	3.27 ± 2.74	1.53 ± 1.50	1.22 ± 5.46	-
CKD	N (%)	N (%)	N (%)	N (%)	N (%)
NO CKD	8 (100)	0	0	0	14 (93)
CKD-1		5 (26.3)	0	0	1 (6.7)
CKD-2		4 (21.1)	3 (8.8)	1 (6.7)	0
CKD-3		2 (10.5)	20 (58.8)	9 (60)	0
CKD-4		6 (31.6)	10 (29.4)	5 (33.3)	0
CKD-5		2 (10.5)	1 (2.9)	0	0
COMORBIDITIES	N (%)	N (%)	N (%)	N (%)	N (%)
AH	4 (50)	12 (63.2)	31 (91.2)	13 (86.6)	6 (40)
DM	0	0	9 (26.5)	5 (33.3)	0
SMOK	0	6 (31.6)	5 (14.7)	1 (6.7)	1 (6.7)
HLP	0	9 (47.4)	13 (38.2)	5 (33.3)	0
CAD	0	0	4 (11.8)	1 (6.7)	0
ATH	0	2 (10.5)	4 (11.8)	13 (86.6)	0
KIDNEY DISEASE	N (%)	N (%)	N (%)	N (%)	N (%)
NB	8 (100)	0	2 (5.8)	2 (13.3)	13 (86.6)
GN	-	17 (89.5)	4 (11.7)	1 (6.6)	-
HTN	-	1 (5.3)	-	-	-
ANCA VS	-	1 (5.3)	-	-	-
TCR	-	-	2 (5.8)	1 (6.6)	-
ABMR	-	-	5 (14.7)	1 (6.6)	-
TCR + ABMR	-	-	4 (11.7)	1 (6.6)	-
BKVAN	-	-	3 (8.8)	3 (20)	-
NC	-	0	14 (41.1)	6 (40)	2 (13.3)
ATI	6 (75)	5 (26.3)	8 (23.5)	3 (20)	8 (53.3)

CKD, chronic kidney disease (defined by KDIGO guidelines (Rf); CTRL, control group samples from kidney donor nephrectomies without any evidence of CKD; eGFR > 90 mL/min, absence of proteinuria or structural abnormalities; NKB, native kidney biopsy group; TXCI, clinically indicated transplant biopsy group; TX0, zero time kidney transplant biopsyfor zero-time biopsies and for CTRL clinical data from kidney donors were used; N, number of subjects; BMI, body mass index; eGFR, estimated glomerular filtration (CKD-EPI formula used); 24-h PROT, 24-h proteinuria in g/24 h urine collection (data for donors not available—NA; all donors negative were negative for protein in urine by dipstick protein trace); CKD stage, chronic kidney disease stage: 0—no CKD (eGFR ≥ 90 mL/min, without kidney damage), 1 (eGFR ≥ 90 + presence of kidney damage); 2 (eGFR 60–90 + presence of kidney damage); 3 (eGFR 30–59); 4 (eGFR 15–29); 5 (eGFR ≤ 15); AH, arterial hypertension; DM, diabetes mellitus; SMOK, smoking; HLP, hyperlipoproteinemia; CAD, coronary artery disease; ATH, any atherosclerotic event (cardiovascular or cerebrovascular or peripheral artery disease); KIDNEY DISEASE, underlying histopathologic diagnosis of kidney disease; NB, normal biopsy; GN, primary glomerular disease; HTN, hypertensive kidney disease; ANCAVS, ANCA vasculitis; TCR, T-cell-mediated rejection; ABMR, antibody mediated rejection; TCR + ABMR, combined rejection; BKVAN, BK virus-associated nephropathy; NC, nonspecific chronic changes without criteria for specific diagnosis; ATI, acute tubular injury.

**Table 2 diagnostics-12-00648-t002:** Paired analysis of ADAMTS-4 expression between clinically indicated transplant biopsies (TXCI) and their belonging zero-time transplant biopsies (TX0)—15 pairs.

	TX0 (N = 15)	TXCI (N = 15)	*p*
eGFR (mean)	91.47	37.93	<0.00001
CKD STAGE (median)	0 (0–3)	3 (2–4)	<0.00001
CI SCORE (mean)	0.07	1.27	0.000157
ADAMTS-4 INT (n/N)	1/15	10/15	0.00169
ADAMTS-4 PTC (n/N)	0/15	15/15	<0.0001
ADAMTS-4 PXT (n/N)	6/15	2/15	NS
ADAMTS-4 DT (n/N)	11/15	15/15	NS
ADAMTS-4 GC (n/N)	5/15	13/15	0.00778
ADAMTS-4 BW (n/N)	14/15	10/15	NS
TIMEPOINT OF BIOPSY (days)	0	690 (28–2340)	

eGFR, estimated glomerular filtration rate (ml/min), calculated by CKD-EPI formula; CKD stage, chronic kidney disease stage: 0—no CKD (eGFR ≥ 90 mL/min, without kidney damage), 1 (eGFR ≥ 90 + presence of kidney damage); 2 (eGFR 60–90 + presence of kidney damage); 3 (eGFR 30–59); 4 (eGFR 15–29); 5 (eGFR ≤ 15); ci score, chronic interstitial score (Banff classification); ADAMTS-4, A disintegrin and metalloproteinase with thrombospondin motifs 4; INT, interstitial stroma; PTC, peritubular capillaries; PXT, proximal tubules; DT, distal tubules; GC, glomerular capillaries; BW, Bowman space; TIMEPOINT OF BIOPSY (days), timing of kidney biopsy procedure (days after kidney transplantation). Data distribution differences were compared using Fisher’s exact test.

## Data Availability

The data presented in this study are available on request from the corresponding author.

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
