# Peer review of "First Characterization of ADAMTS-4 in Kidney Tissue and Plasma of Patients with Chronic Kidney Disease—A Potential Novel Diagnostic Indicator"

_diagnostics, 2022, doi:10.3390/diagnostics12030648_

Round 1

Reviewer 1 Report

In this study the authors have characterized ADAMTS-4 in kidney tissues and plasma of patients with various stages of chronic kidney disease (CKD) to evaluate the opportunity of its use as a novel biomarker of this disease.

To improve the present manuscript the following specific comments should be addressed:

Introduction: The authors have based this study on their previous results stating that metzicin protease family member ADAMTS-4 (also known as aggrecanase-1) appeared in the peripheral blood of human patients with ESRD [reference 13]. However, this reference is incomplete in the bibliographical list, and I could not find in the available literature. Moreover, it is not new since the only specified element is the year of publication (2012). Thus, they should provide the details of this previously published paper.

Results and Figures: The number of plasma samples from patients with CKD and analyzed by ELISA should be provided both in the text and in Figure 1. How many samples did they analyze for each of the 8 patient groups?

Discussion: Have the authors analyzed other members of this protein family, such as ADAMTS-1 or ADAMTS-13? Besides, they should discuss literature data on the possible involvement of the other related proteins in CKD.

Author Response

In this study the authors have characterized ADAMTS-4 in kidney tissues and plasma of patients with various stages of chronic kidney disease (CKD) to evaluate the opportunity of its use as a novel biomarker of this disease.

To improve the present manuscript the following specific comments should be addressed:

Introduction: The authors have based this study on their previous results stating that metzicin protease family member ADAMTS-4 (also known as aggrecanase-1) appeared in the peripheral blood of human patients with ESRD [reference 13]. However, this reference is incomplete in the bibliographical list, and I could not find in the available literature. Moreover, it is not new since the only specified element is the year of publication (2012). Thus, they should provide the details of this previously published paper.

We thank the Reviewer for their remark. The reference in question is not a reference of a manuscript, but a reference of the patent (the corresponding author is one of the inventors). We thank the Reviewer – we highlihgted the fact that it is a patent, not a manuscript in the references. The patent details can be accessed on the following link: https://patents.google.com/patent/US8263072B2/en

The reference in question now reads:

[13]      Grgurevic L, Vukicevic S. ADAMTS4 as a blood biomarker and therapeutic target for chronic renal failure. 2012. (Patent No. US 8,263,072 B2)

Results and Figures: The number of plasma samples from patients with CKD and analyzed by ELISA should be provided both in the text and in Figure 1. How many samples did they analyze for each of the 8 patient groups?

We thank the Reviewer for their remark and apologize for making this omission. This is now corrected in the revised Figure 1 and in a paragraph of the Materials and Methods section, which reads: „Experimental groups included: 32 patients in CKD stages 1-5, 5 hemodialysis (HD), 7 peritoneal dialysis (PD) patients, 30 kidney transplant patients (TX) and a control group of 5 healthy volunteers (CTRLpl) (Figure 1).”..

Discussion: Have the authors analyzed other members of this protein family, such as ADAMTS-1 or ADAMTS-13? Besides, they should discuss literature data on the possible involvement of the other related proteins in CKD.

We thank the Reviewer for their remark. We did not analyze ADAMTS-1 or ADAMTS-13. Our work focused on characterization of ADAMTS-4 (the basis of which was the patent invented by the corresponding author) and BMP-1.

Prompted by the Reviewer, we have now included other ADAMTS proteins in the discussion by adding the following: „Other ADAMTS proteins have previously been implicated in the pathogenesis of renal insufficiency: Higher expression of ADAMTS-1 was detected in the early course of fi-brogenesis on mouse model of unilateral ureteral obstruction. Investigators defined endothelial cell of PTC as the source of ADAMTS-1 in the process of pericytes detach-ment and PTC destabilization [25]. ADAMTS-2 and -12 were upregulated in mesen-chymal interstitial cells during their transformation to myofibroblasts [26]. Further-more, ADAMTS-1, -12 and -15 were found upregulated in animal model of adriamycin induced kidney failure [27].“

Reviewer 2 Report

Suggestions are marked in the attached file.

Author Response

Suggestions are marked in the attached file. Do fibroblast cells from normal healthy controls go back to resting state in vitro? Probably not, as they lack the environment. We thank the Reviewer for their remark. Upon further consideration and research, we agree with the Reviewer and have thus decided to remove the mentioned section from that sentence which now reads: „In so doing, they express α-smooth muscle actin, indicating their activation.“ I would also look at the paper below for clues: Boyd, David F., et al. "Exuberant fibroblast activity compromises lung function via ADAMTS4." Nature 587.7834 (2020): 466-471. We thank the Reviewer for pointing us towards this paper as it helps su complete this section of the Discussion. We added the following sentence, along with the reference: „Boyd et al. showed that fibroblasts produce ADAMTS-4 in response to tissue-damage which leads to ECM remodeling and promotion of immune cell infiltration in the lungs [28].“

Reviewer 3 Report

The authors have identified ADAMTS-4 as a diagnostic indicator in kidney tissues and 369 plasma of patients with CKD. The authors could get significant results although the sample size is relatively small. However, the manuscript can be improved in some aspects.

  1. Too many abbreviations and P values reduced the readability of the abstract. The abstract should be modified.
  2. Results of statistical analysis should be included in the figures.
  3.  Some typos and grammatical errors should be corrected.

Author Response

The authors have identified ADAMTS-4 as a diagnostic indicator in kidney tissues and plasma of patients with CKD. The authors could get significant results although the sample size is relatively small. However, the manuscript can be improved in some aspects.

  1. Too many abbreviations and P values reduced the readability of the abstract. The abstract should be modified.

We thank the Reviewer for helping us improve the quality of our manuscript. We have now modified the abstract in order to make it more comprehensible..

  1. Results of statistical analysis should be included in the figures.

We thank the Reviewer for their pointing this out. We have revised Figure 2 and Figure 5 in order to present the results of statistical analysis. Furthermore, prompted by the Reviewer's remark that results must be presented more clearly – we have revised the results section.

  1.  Some typos and grammatical errors should be corrected.

We thank the Reviewer for their remarks. We carefully proof-read the manuscript and improved grammar and spelling in the mistakes we found.

Round 2

Reviewer 1 Report

No further comments